# A novel gene *ZNF862* causes hereditary gingival fibromatosis

**Juan Wu[1†], Dongna Chen[2†], Hui Huang[2†], Ning Luo[1], Huishuang Chen[2], Junjie Zhao[1], Yanyan Wang[2], Tian Zhao[1], Siyuan Huang[2,3], Yang Ren[1], Teng Zhai[2], Weibin Sun[1], Houxuan Li[1]\*, Wei Li[2]\***

[1]Department of Periodontology, Nanjing Stomatological Hospital, Medical School of Nanjing University, Nanjing, China; [2]BGI Genomics, BGI-Shenzhen, Shenzhen, China; [3]Academy for Advanced Interdisciplinary Studies, Peking University, Beijing, China

**Abstract** Hereditary gingival fibromatosis (HGF) is the most common genetic form of gingival fibromatosis which is featured as a localized or generalized overgrowth of gingivae. Currently two genes (*SOS1* and *REST*), as well as four loci (2p22.1, 2p23.3–p22.3, 5q13–q22, and 11p15), have been identified as associated with HGF in a dominant inheritance pattern. Here, we report 13 individuals with autosomal-dominant HGF from a four-generation Chinese family. Whole-exome sequencing followed by further genetic co-segregation analysis was performed for the family members across three generations. A novel heterozygous missense mutation (c.2812G > A) in zinc finger protein 862 gene (*ZNF862*) was identified, and it is absent among the population as per the Genome Aggregation Database. The functional study supports a biological role of *ZNF862* for increasing the profibrotic factors particularly COL1A1 synthesis and hence resulting in HGF. Here, for the first time we identify the physiological role of *ZNF862* for the association with the HGF.

**\*For correspondence:**
lihouxuan3435_0@163.com (HL);
chorway@pku.edu.cn (WL)

†These authors contributed equally to this work

## Editor's evaluation

This study is of clinical relevance to those interested in the etiology and pathology of hereditary gingival fibromatosis (HGF). The paper discusses two novel findings: identification of a causative role of a missense mutation in the gene encoding the zinc finger protein 862 (ZNF862) that leads to hereditary gingival fibromatosis (HGF), a rare disease characterized by overgrowth of gingivae, in an examined family, and a suggestion of the molecular consequences of that mutation that leads to the disease.

## Introduction

Gingival fibromatosis (GF) is a rare disorder which is characterized by a benign, nonhemorrhagic, localized, or generalized fibrous enlargement of free and attached gingivae with slow progression. GF could be complicated by epilepsy, hypertrichosis, and mental retardation (*Balaji and Balaji, 2017*; *Snyder, 1965*) or it can develop as a part of syndromes like Cowden's syndrome (*Witkop, 1971*), Zimmerman-Laband syndrome (*Guglielmi et al., 2019*), Cross syndrome (*Poulopoulos et al., 2011*), Rutherford syndrome (*Häkkinen and Csiszar, 2007*), Ramon syndrome (*Suhanya et al., 2010*), Jones syndrome (*Gita et al., 2014*), Costello syndrome (*Hennekam, 2003*), Ectro-amelia syndrome (*Morey and Higgins, 1990*), and hyaline fibromatosis syndrome (*Hamada et al., 1980*); it may also be caused by poor oral hygiene-related gingival inflammation, puberty, pregnancy, or a side effect of the common medications such as calcium channel blockers (nifedipine and verapamil) (*Livada and Shiloah, 2012*); usually it betrays as an isolated condition as non-syndromic hereditary gingival fibromatosis (HGF) (*Jorgenson and Cocker, 1974*). HGF is the most common genetic form of GF and usually transmitted

as an autosomal-dominant inheritance pattern. Nevertheless, sporadic cases and autosomal-recessive inheritance pedigrees have also been previously described (*Majumder et al., 2013*).

As reported previously, to date, four loci (2p22.1 [MIM: 135,300], 5q13–q22 [MIM: 605,544], 2p23.3–p22.3 [MIM: 609,955], and 11p15 [MIM: 611,010]) have been identified associated with HGF (*Xiao et al., 2001*; *Ye et al., 2005*; *Zhu et al., 2007*; *Xiao et al., 2000*); besides a heterozygous frame-shift mutation in *SOS1* (MIM: 182530) has been identified as causative for the autosomal-dominant HGF in a Brazilian family (*Hart et al., 2002*); and protein truncating mutations in *REST* (MIM: 600571) have been reported as the genetic cause of an autosomal-dominant inheritance pattern HGF across three independent families (*Bayram et al., 2017*). The estimated incidence of HGF is 1 per 175,000 of the population (*Ahmed and Ali, 2015*), and equal between males and females (*Gawron et al., 2016*; *Odessey et al., 2006*). Periodontal surgery including gingivectomy, gingivoplasty, and flap surgery can be applied to patients with HGF due to the aesthetic and functional concern, whereas the recurrence of hyperplasia is relatively high potentially owing to the underlying genetic predisposition (*Chaurasia, 2014*; *Zhou et al., 2011*). Therefore, it is important to explore the genetic causes and etiology accounting for HGF with expectations of the precise genetic diagnosis and of the potential gene therapy development, to help individuals who desire to circumvent the sufferings and improve their lives. We report a large Chinese family with 13 affected individuals clinically diagnosed with non-syndromic HGF, and with 12 unaffected family members. Genetic analysis of this large family led to the identification of a heterozygous mutations in a novel gene *ZNF862*.

## Results and discussion

This four-generation pedigree with HGF was shown in *Figure 1A*. Visual inspection indicated severe or mild generalized GF in all affected members, whereas not in any unaffected member, from this family. No individuals in this family was ever exposed to the medication that may result in GF according to our investigation. No other congenital abnormality has been found for all family members. The clinical data of all individuals in this family are summarized in *Table 1*, and clinical photos of one control and three patients from each generation are available in *Figure 1B*. From a clinical perspective, the proband (IV-2) had the mild recurrence of hyperplasia during the second year after the periodontal surgery of gingivectomy, gingivoplasty, and flap surgery; while the patient II-2 had an additional clinical finding (hypertension, with no medication of calcium channel blockers treated).

Whole-exome sequencing (WES) was performed for the proband and nine other members (numbers indicated in red color in *Figure 1A*) in the family according to previously described protocols (*Zhang et al., 2015*). After analyzing all single nucleotide variants (SNVs) and indels for each gene including the promoter region, exons, splicing sites, introns, and untranslated region (UTR), only three variants *ATP7B* (c.3403G > A), *CDADC1* (c.83–13G > T), *ZNF862* (c.2812G > A) were identified to be co-segregated with the phenotype among the 10 family members who underwent WES. To further screen and validate the potential disease-causing variants co-segregation with the phenotype in this pedigree, conventional PCR was performed for sanger sequencing evaluation. Eventually, only *ZNF862* (c.2812G > A), no any other variant, was validated to be co-segregated with the phenotype in all 23 alive members in this family, as is shown in *Figure 1C*.

*ZNF862* gene harbors eight exons, encodes a 1169-amino acid protein, and is located on chromosome 7q36.1. It does not escape our notice that *ZNF862* does not map to any one of the loci that were reported previously as be associated with HGF. ZNF862 is a predicted intracellular protein which function, to the best of our knowledge, is not yet clearly identified, as a zinc finger protein it may be involved in transcriptional regulation. ZNF862 is expressed ubiquitously across tissues (*Fagerberg et al., 2014*), it may play various roles under different physiological condition. *Schwartz et al., 2019*, suggested a plausible role of ZNF862 in matrix-producing metaplastic carcinoma. *Peng et al., 2018*, reported ZNF862 associated with IgE-mediated type-I hypersensitivity in children. The WES-identified heterozygous variant (p.A938T) in our study located close to and upstream of the Dimer_Tnp_hAT (hAT family C-terminal dimerization region) domain of ZNF862 protein (*Figure 1*).

The pathophysiological mechanisms underlying HGF remain largely elusive. Nevertheless, over-production of extracellular matrix, particularly the major component collagen type I (COL1A1), may account for the gingival fibroblast overgrowth phenotype; meanwhile increased synthesis of TIMP-1 seems to be associated with COL1A1 excessive accumulation in HGF gingival fibroblasts (*Gawron et al., 2018*; *Roman-Malo et al., 2019*). Besides, Martelli-Junior et al. reported that overexpression

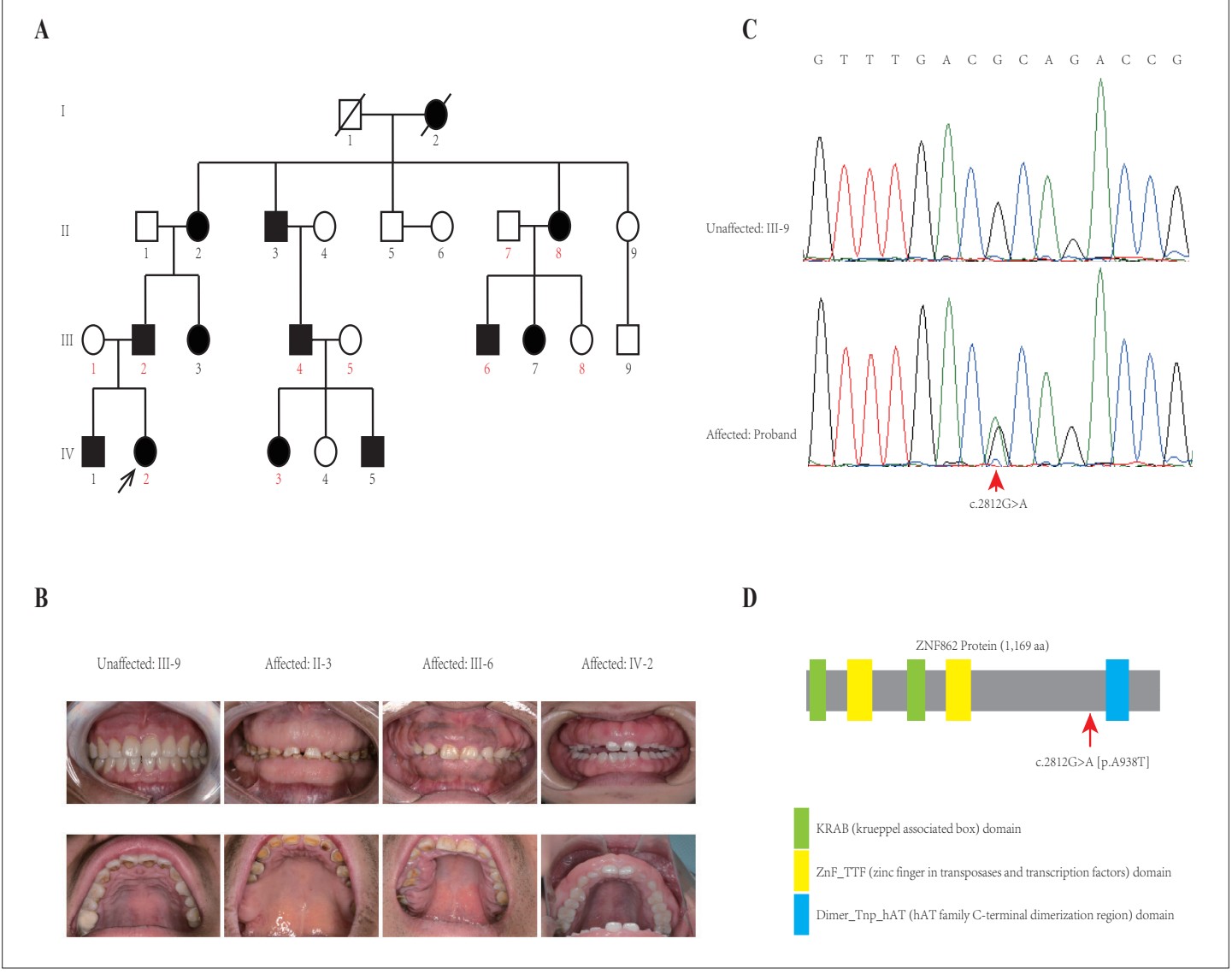

**Figure 1.** Pedigree and co-segregation analysis. (**A**) The black arrow indicates the proband. The affected individuals are indicated with black-filled boxes in this family. Whole-exome sequencing (WES) was performed for the proband and nine other members (numbers indicated in red color). (**B**) The photographs of gingival overgrowth are revealed by intraoral examination in affected members compared with in the unaffected member in this family. (**C**) In this family, all affected individuals harbor the heterozygous variant (c.2812G > A) whereas the unaffected individuals are wild-type. (**D**) Schematic structure with putative domains of ZNF862 protein and the localization of the novel variant (red arrows). Green rectangles indicate KRAB (krueppel-associated box) domains; yellow rectangles indicate ZnF_TTF (zinc finger in transposases and transcription factors) domains; blue rectangle indicates Dimer_Tnp_hAT (hAT family C-terminal dimerization region) domain.

of TGF-β1 and IL-6 in gingival fibroblasts play pivotal roles in increasing the synthesis of the COL1A1 along with other specific growth factors (*Martelli-Junior et al., 2003*). Systematically, transcriptomic analysis of HGF patients and controls performed by *Han et al., 2019*, demonstrated that regulatory network connection of TGF-β/SMAD signaling pathway and craniofacial development processes contributes to the molecular mechanism of clinical-pathological manifestations.

Aiming to explore the underlying mechanism related to the *ZNF862* mutation, RNA sequencing was performed according to previously described protocols (*Liang et al., 2019*). The primary fibroblasts were isolated from fibromatic gingival specimens obtained from two patients who underwent gingivectomy (IV-1 and IV-2) using standard explant culture as described previously (*de Andrade et al., 2001*), while control cells were obtained from four independent age- and gender-matched controls who underwent crown lengthening surgery for the restorative purpose. The genes expression profiling

**Table 1.** Clinical Characteristics of the individuals in this family.
NA = Not available.

| Members | Identified variant | Age at last exam | Gender | Gingival fibromatosis | Age of onset | Exposure to medication | Gingivectomy/recurrence | Other clinical finding |
|---|---|---|---|---|---|---|---|---|
| II–1 | Wild type | 67 years | Male | – | – | No | -/- | No |
| II–2 | p.A938T (ZNF862) | 70 years | Female | + | NA | No | -/- | Hypertension |
| II-3 | p.A938T (ZNF862) | 66 years | Male | + | NA | No | -/- | No |
| II-4 | Wild type | 66 years | Female | – | – | No | -/- | No |
| II-5 | Wild type | 64 years | Male | – | – | No | -/- | No |
| II-6 | Wild type | 64 years | Female | – | – | No | -/- | No |
| II –7 | Wild type | 62 years | Male | – | – | No | -/- | No |
| II –8 | p.A938T (ZNF862) | 60 years | Female | + | NA | No | -/- | No |
| II –9 | Wild type | 54 years | Female | – | – | No | -/- | No |
| III–1 | Wild type | 49 years | Female | – | – | No | -/- | No |
| III–2 | p.A938T (ZNF862) | 47 years | Male | + | NA | No | -/- | No |
| III–3 | p.A938T (ZNF862) | 45 years | Female | + (mild) | NA | No | -/- | No |
| III–4 | p.A938T (ZNF862) | 41 years | Male | + | NA | No | -/- | No |
| III–5 | Wild type | 36 years | Female | – | – | No | -/- | No |
| III–6 | p.A938T (ZNF862) | 36 years | Male | + | NA | No | -/- | No |
| III–7 | p.A938T (ZNF862) | 34 years | Female | + | NA | No | -/- | No |
| III–8 | Wild type | 32 years | Female | – | – | No | -/- | No |
| III–9 | Wild type | 30 years | Male | – | – | No | -/- | No |
| IV–1 | p.A938T (ZNF862) | 26 years | Male | + | 6–7 years | No | +/- | No |
| IV–2 | p.A938T (ZNF862) | 22 years | Female | + | 6–7 years | No | +/+ (mild) | No |
| IV–3 | p.A938T (ZNF862) | 13 years | Female | + | 2–3 years | No | -/- | No |

*Table 1 continued on next page*

*Table 1 continued*

| Members | Identified variant | Age at last exam | Gender | Gingival fibromatosis | Age of onset | Exposure to medication | Gingivectomy/ recurrence | Other clinical finding |
|---------|--------------------|-------------------|--------|-----------------------|--------------|------------------------|--------------------------|-----------------------|
| IV-4 | Wild type | 11 years | Female | – | – | No | -/- | No |
| IV-5 | p.A938T (ZNF862) | 7 years | Male | + | 2–3 years | No | -/- | No |

and changes in patients over controls were summarized in *Supplementary file 1*, the samples average counts per million mapped reads (CPM) for each gene is shown, the expression fold-change (FC) and their statistical significance false discovery rate (FDR) are also provided. The expression profiling of attested profibrotic genes (*Gao et al., 2019*), which may be involved in HGF, was interrogated (*Figure 2A*). A part of such genes, including *COL1A1*, *TGFB1/2,* and *SMAD1*, was up-regulated in HGF fibroblasts compared with controls; whereas another part was down-regulated, suggesting that the special HGF in our study was attributed to a corresponding part of profibrotic factors, including TGF-β/SMAD1 signaling pathway and COL1A1. Besides RNA sequencing, the in vitro cultured gingival fibroblasts from control 1 underwent adenovirus-mediated delivery of short hairpin RNA (shRNA) targeting *ZNF862*. The expression of four most significant genes *ACTA2*, *FOS*, *SMAD1*, *AGT*, as well as *COL1A1*, under shRNA treatment were investigated (*Figure 2B*), the knockdown of *ZNF862* in control gingival fibroblasts leads to the profibrotic genes up-regulation, reminiscent of the expression in the patients. Here, we speculate that the *ZNF862* mutation in patients attenuate the protein function, similar with the *ZNF862* shRNA, and promote special profibrotic genes expression to result in the HGF trait. Meanwhile, it did not escape our notice that the causative gene *SOS1* was mildly down-regulated, which may be associated with the phenotype (*Figure 2* and *Figure 2—figure supplement 3*). Nevertheless, the proliferation in control gingival fibroblasts was not significantly stimulated by the knockdown of *ZNF862* during 5 days' culture period (*Figure 2—figure supplement 2*), which may be partially resulting from the too short time course to observe the sophisticated physiological effect.

To screen all the genes differentially expressed (DE) between patients and controls, we quantified genes whose expression changed over twofold ($|\log_2 FC| \geq 1.0$), and we employed a cut-off of FDR < 0.05 due to the statistical power limitation of the small number of entire samples (N = 6). Finally, the transcriptomic comparison of HGF patients and controls yielded 597 DE transcripts, 355 up-regulated while 242 down-regulated (Figure S1). Of which, the 100 most up-regulated DE genes and 100 most down-regulated DE genes (*Supplementary file 2*) were shown in *Figure 2C*, furthermore the top 10 genes were illustrated accordingly. These DE genes probably correlated with the HGF trait. The 20 most significantly enriched functional pathways of all DE genes were shown (*Figure 2D*). Thereinto, lipoic acid metabolism is the most enriched cluster; notably the MAPK signaling pathway is associated with IL-6 pathway, besides several infection-related clusters in this chart is related with IL-6-induced autophagy pathway.

Taken together we can speculate that ZNF862 may execute as transcriptional repressors since ZNF862 has the zinc-finger DNA-binding domain, meanwhile this missense mutation may attenuate biological function of ZNF862 and hereby enhance the TGF-β/SMAD1 signaling pathway that increases profibrotic factors, particularly COL1A1 accumulation in gingival tissue and results in HGF. Nevertheless, the underlying mechanism of this novel mutation leading to HGF remains enigmatic, with no more specific experiment designed for exploring the biological function of *ZNF862*. Further studies are expected to expand the mutational spectrum of *ZNF862* and the associations with phenotypes, and to investigate the physiological role of *ZNF862* in the process of HGF.

In this study, we identify a missense variant (c.2812G > A) in a novel gene *ZNF862* that causes autosomal-dominant HGF in a four-generation Chinese family. The functional study supports a biological role of *ZNF862* for increasing the profibrotic factors, particularly COL1A1 synthesis in gingiva and hence resulting in HGF. This variant has been absent among the large population according to the report of Exome Aggregation Consortium (ExAC) database, 1000 Genomes, and Genome Aggregation Database. The log of the odds (LOD) score is calculated as 3.6 based on the dominant segregations. According to the combining criteria set guidelines by the American College of Medical Genetics and Genomics (ACMG) and the Clinical Genome Resource (ClinGen) (*Oza et al., 2018*; *Richards et al., 2015*), the missense variant is rated as 'Pathogenic' with the following evidence rules: PP1_Strong (LOD > 3), PS3 (in vitro functional study support), and PM2 (absent from controls). We can speculate that this finding may shed lights on the precise genetic diagnosis and on the potential therapy development, helping individuals continue their lives better.

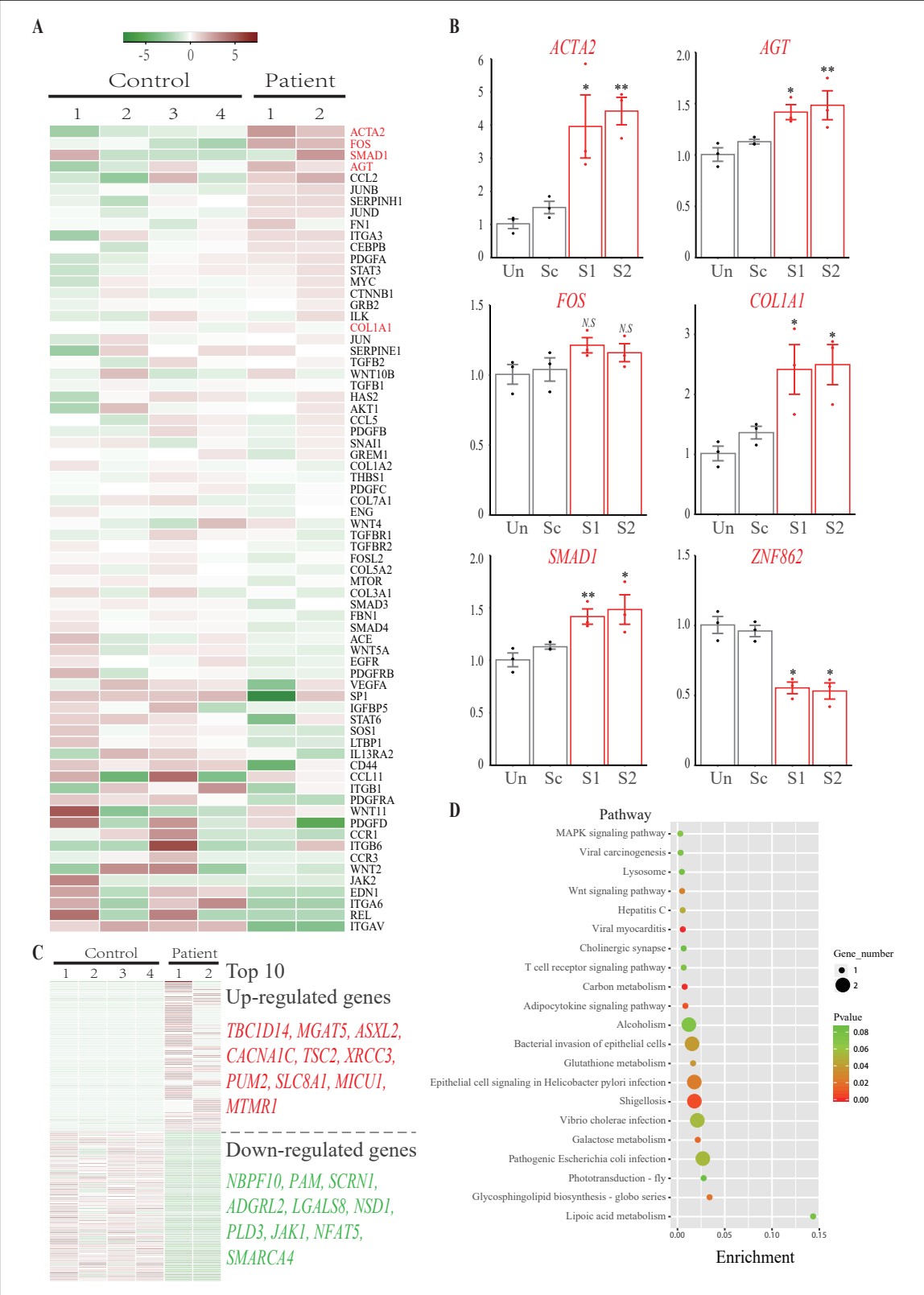

**Figure 2.** RNA sequencing and knockdown analysis. (**A**) Heatmap of the 70 attested profibrotic genes expression profiling in RNA sequencing. Rows represent genes and columns represent samples. The heatmap is color-coded based on an average normalization; red represents high expression value and green represents low expression value. (**B**) Real-time PCR analysis of *ACTA2*, *FOS*, *SMAD1*, *AGT*, *COL1A1*, and *ZNF862* mRNA abundance in untreated (Un), adenovirus delivered scrambler (Sc), adenovirus delivered *ZNF862* short hairpin RNA (shRNA) 1 (**S1**), and adenovirus delivered *ZNF862*

*Figure 2 continued*

shRNA 2 (**S2**) groups, respectively. Results are given as means ± SEM, dots indicate the relative values of gene expression levels. *p < 0.05, **p < 0.01 in comparison with Sc group. *N.S.* means not significant. (**C**) Heatmap of the 100 most up-regulated differentially expressed (DE) genes and 100 most down-regulated DE genes expression profiling in RNA sequencing. The color key is the same with (**A**). (**D**) Functional annotation chart of DE genes, the 20 most significantly enriched functions of DE genes were plotted. Functional pathways enrichment tests were based on KEGG database. Enrichment denotes the proportion of DE genes among the indicated pathway.

The online version of this article includes the following figure supplement(s) for figure 2:

**Figure supplement 1.** Transcriptome comparison between hereditary gingival fibromatosis (HGF) patients and controls.

**Figure supplement 2.** The proliferation of gingival fibroblasts.

**Figure supplement 3.** SOS1 expression in gingival fibroblasts.

## Materials and methods

### Samples collection and WES

The participants from this family were recruited from Nanjing Stomatological Hospital. The genomic DNA was extracted from the peripheral blood. WES was performed for probands and nine other members (numbers indicated in red color in *Figure 1A*) in the family. Average WES sequencing depth of target region for each sample was higher than 100, and 20× coverage for more than 95% targeted bases was achieved for each sample. Sequencing reads were aligned to the human genome reference sequence (hg19) through the in-house programs. Genome Analysis Toolkit (GATK) was used to call SNVs and indels; and Annotation of Genetic Variants (ANNOVAR) was used for the annotation. A custom Perl script (*Source code 1*) was used for retrieving the variants corresponding to the inheritance model of this family and that do not appear in unaffected individuals. All the WES data underwent the same quality control filtering and pruning procedures to maximize parity. The following primers are used to amplify products for Sanger sequencing: *ZNF862* (NM_001099220.3) forward: 5'-TCATGCGCGGCT TCCACTTTGT; reverse: 5'-AGCACTCGAAATACCTGGCCAG; *ATP7B* (NM_000053.4) forward: 5'-TCCAGTCGGTAACCTGTTCAC; reverse: 5'-GCACAGCAGAGGCAATCAC; *CDADC1* (NM_030911.4) forward: 5'-CGACTAAGGCTGAAGCGTCT; reverse: 5'-AATTGGGGGAAATTATGTGG.

### Transcriptome sequencing

The primary fibroblasts were isolated from fibromatic gingival specimens obtained from the patients who underwent gingivectomy (IV-1 as patient 1 and IV-2 as patient 2) using standard explant culture, while control cells were obtained from independent age- and gender-matched controls (two males at 25 and 26 years of age as controls 1 and 2, respectively, and two females at 21 and 23 years of age as controls 3 and 4, respectively) who underwent crown lengthening surgery for the restorative purpose.

Total RNA samples were extracted from the cultured fibroblast cells using Trizol, RNA integrity was assessed using the RNA Nano 6000 Assay Kit of the Bioanalyzer 2100 system (Agilent Technologies, CA). A total amount of 1 µg RNA per sample was used as input material for the RNA sample preparations. Briefly, mRNA was purified from total RNA using poly-T oligo-attached magnetic beads. Fragmentation was carried out using divalent cations under elevated temperature in First Strand Synthesis Reaction Buffer. First strand cDNA was synthesized using random hexamer primer and M-MuLV Reverse Transcriptase (RNase H-). Second strand cDNA synthesis was subsequently performed using DNA Polymerase I and RNase H. Remaining overhangs were converted into blunt ends via exonuclease/polymerase activities. After adenylation of 3' ends of DNA fragments, adaptor with hairpin loop structure was ligated to prepare for hybridization. In order to select cDNA fragments of preferentially 370–420 bp in length, the library fragments were purified with AMPure XP system (Beckman Coulter, Beverly, MA). Then PCR was performed with Phusion High-Fidelity DNA polymerase, Universal PCR primers, and Index Primer. At last, PCR products were purified (AMPure XP system) and library quality was assessed on the Agilent Bioanalyzer 2100 system. The library preparations were sequenced on an Illumina Novaseq platform and 150 bp paired-end reads were generated.

Raw data were processed through an in-house bioinformatics workflow. Paired-end clean reads were aligned to the reference genome ensembl release-97 (http://asia.ensembl.org/Homo_sapiens/Info/Index) using Hisat2 v2.0.5, featureCounts v1.5.0-p3 was used to count the reads numbers mapped to each gene. And then the CPM and fragments per kilobase of transcript per million mapped reads (FPKM) of each gene was calculated based on the length of the gene and reads count mapped to

this gene. At least 45 million clean mapped reads per sample were yielded. The clean bases after filtration for each sample are higher than 6.7 Giga-base, with the effective rate higher than 99.7%, the Q20 bases is higher than 94% for each sample. The GC content for each sample is ranging from 51% to 55%. DE analysis of two groups (patients and controls) was performed using the edgeR R package (3.22.5). The resulting p-values were adjusted using the Benjamini and Hochberg's approach for controlling the FDR. Corrected p-value of 0.05 and absolute FC of 2 were set as the threshold for significantly DE. We used clusterProfiler R package to test the statistical enrichment of DE genes in KEGG pathways, corrected p-value less than 0.05 was considered significantly enriched. KEGG is a database resource for understanding high-level functions and utilities of the biological system (http://www.genome.jp/kegg/).

## ShRNA interference

Besides RNA sequencing, the in vitro cultured fibroblasts (from control 1, control 2, and control 3, respectively) underwent adenovirus-mediated delivery of shRNA targeting *ZNF862*, the following sequences were used for the *ZNF862* shRNA interference experiments, *ZNF862* shRNA 1: GCTC TGTTCTGCTCTGCTTGC; *ZNF862* shRNA 2: GGATTTACATCCGCTACTTCA; scrambler: GCACCCAG TCCGCCCTGAGCAAA, at 72 hr after the adenovirus-mediated U6 promoter-driven shRNA delivered into the cells (MOI = 70) real-time PCR was performed using 18s rRNA as an internal control. Cells between the third and sixth passages were used for all abovementioned experiments. The following primers are used to amplify products for the real-time PCR: *FOS* (NM_005252) forward: 5'-GCCG GGGATAGCCTCTCTTACT; reverse: 5'-CCAGGTCCGTGCAGAAGTC; *SMAD1* (NM_001003688) forward: 5'-AGAGACTTCTTGGGTGGAAACA; reverse: 5'-ATGGTGACACAGTTACTCGGT; *AGT* (NM_000029) forward: 5'-GCTGACAGGCTACAGGCAATC; reverse: 5'-TGTGAACACGCCCACCACC; *CCL2* (NM_002982) forward: 5'-CAGCCAGATGCAATCAATGCC; reverse: 5'-TGGAATCCTGAACCCA CTTCT; *COL1A1* (NM_000088.3) forward: 5'-GAGTCTACATGTCTAGGGTCT; reverse: 5'-CACGTCATC-GCACAACACCT; *ZNF862* (NM_001099220.3) forward: 5'-CTTACTCCAGGAGGAATGGGT; reverse: 5'-CTCCCATGTAGCCCATCTGT; *ACTA2* (NM_001613.3) forward: 5'-CCAGACATCAGGGGGTGAT; reverse: 5'-TGGTGCCAGATCTTTTCCAT; *FN1* (NM_001365524.2) forward: 5'-CCATAAAGGGCAAC-CAAGAG; reverse: 5'-AAACCAATTCTTGGAGCAGG; *SOS1* (NM_005633.4) forward: 5'- GAAACCCT TTATCTCTCCCAGT; reverse: 5'- CTTGTCAGCACACATTGCCACT; *18srRNA* (NR_145822.1) forward: 5'-CGAACGTCTGCCCTATCAACT; reverse: 5'-CAGACTTGCCCTCCAATGGATCCTCGTT.

## Proliferation assay

Cell Counting Kit 8 (CCK8, Dojindo Molecular Technologies) proliferation assay was performed with the above-mentioned in vitro cultured fibroblasts. The cells in the logarithmic growth phase were plated overnight in a 96-well plate at an initial density of 2000 per well. Starting with adenovirus infection (MOI = 70), 10 µl CCK8 per well was added to the culture medium on the days 1, 2, 3, 4, and 5. The fluorescent optical density value was measured using a microplate reader. The in vitro cultured fibroblasts underwent adenovirus-mediated delivery of *ZNF862* siRNA (NM_001099220.3).

## Statistical analysis

All data from the semi-quantitative real-time PCR analyses are representative of at least three independent experiments. Data are presented as the mean ± SEM of at least three independent experiments and differences (in comparison with Sc group) are considered statistically significant at p < 0.05 using Student's *t*-test. *p < 0.05, **p < 0.01. *N.S.* means not significant. All analyses were performed using R v4.0.2.

## Web resources

1000 Genomes, http://www.internationalgenome.org/
ClinVar, https://www.ncbi.nlm.nih.gov/clinvar/
ExAC, http://exac.broadinstitute.org
Genome Aggregation Database, https://gnomad.broadinstitute.org/
GenBank, http://www.ncbi.nlm.nih.gov/genbank/
OMIM, http://www.omim.org/
UniProt, http://www.uniprot.org/

## Acknowledgements

We thank all the participants in this study, and the support by the funding from National Natural Science Foundation of China (51772144); the Medical Science and Technology Development Foundation, Nanjing Department of Health (YKK18121); Nanjing Clinical Research Center for Oral Diseases, Nanjing Department of Health (No. 2019060009); Jiangsu Province Natural Science Foundation of China (BK20200149).

## Additional information

### Competing interests

Dongna Chen, Hui Huang, Huishuang Chen, Yanyan Wang, Teng Zhai, Wei Li: is employee of BGI Genomics. The other authors declare that no competing interests exist.

### Funding

| Funder | Grant reference number | Author |
| --- | --- | --- |
| National Natural Science Foundation of China | 51772144 | Houxuan Li |
| Medical Science and Technology Development Foundation, Nanjing Department of Health | YKK20152 | Juan Wu |
| Nanjing Clinical Research Center for Oral Diseases, Nanjing Department of Health | 2019060009 | Juan Wu |
| Jiangsu Province Natural Science Foundation of China | BK20200149 | Juan Wu |

The funders had no role in study design, data collection and interpretation, or the decision to submit the work for publication.

### Author contributions

Juan Wu, Investigation, Resources, Validation, Writing – review and editing; Dongna Chen, Formal analysis, Resources, Validation; Hui Huang, Resources, Validation; Ning Luo, Data curation; Huishuang Chen, Data curation, Methodology, Validation, Writing – review and editing; Junjie Zhao, Formal analysis, Methodology; Yanyan Wang, Formal analysis, Methodology, Writing – review and editing; Tian Zhao, Data curation, Investigation; Siyuan Huang, Data curation, Formal analysis, Methodology, Software; Yang Ren, Weibin Sun, Investigation; Teng Zhai, Resources; Houxuan Li, Conceptualization, Funding acquisition, Project administration, Resources, Writing – review and editing; Wei Li, Conceptualization, Data curation, Formal analysis, Project administration, Supervision, Visualization, Writing – original draft, Writing – review and editing

### Author ORCIDs

Juan Wu http://orcid.org/0000-0003-4494-6570
Dongna Chen http://orcid.org/0000-0002-7957-8379
Wei Li http://orcid.org/0000-0003-4475-531X

### Ethics

Human subjects: The usage and handling of human samples in this study was approved by the Institutional Review Board on Bioethics and Biosafety of BGI (IRB No. 19059) and the written informed consent obtained from each participant. Clinical investigation was performed in accordance with the Declaration of Helsinki.

### Decision letter and Author response

Decision letter https://doi.org/10.7554/eLife.66646.sa1

Author response https://doi.org/10.7554/eLife.66646.sa2

## Additional files

### Supplementary files
• Supplementary file 1. The genes expression profiling and changes in patients over controls based on RNA sequencing.
• Supplementary file 2. A list of the 100 most up-regulated differentially expressed (DE) genes and 100 most down-regulated DE genes.
• Transparent reporting form
• Source code 1. Perl Code.

### Data availability
The sequencing data supporting this study have been deposited in the China Genebank Nucleotide Sequence Archive (https://db.cngb.org/cnsa, accession number CNP0000995).

The following dataset was generated:

| Author(s) | Year | Dataset title | Dataset URL | Database and Identifier |
|---|---|---|---|---|
| Wu J, Chen D, Huang H, Luo N, Chen H, Zhao J, Wang Y, Zhao T, Huang S, Ren Y, Zhai T, Sun W, Li W | 2021 | Study on the causative gene of hereditary gingival fibromatosis | https://doi.org/10.26036/CNP0000995 | China National GeneBank, 10.26036/CNP0000995 |

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
