## [Editor Report]

This study is of clinical relevance to those interested in the etiology and pathology of hereditary gingival fibromatosis (HGF). The paper discusses two novel findings: identification of a causative role of a missense mutation in the gene encoding the zinc finger protein 862 (ZNF862) that leads to hereditary gingival fibromatosis (HGF), a rare disease characterized by overgrowth of gingivae, in an examined family, and a suggestion of the molecular consequences of that mutation that leads to the disease.

---

## [Decision Letter]

**Decision letter after peer review:**

Thank you for submitting your article "A Novel Gene ZNF862 Causes Hereditary Gingival Fibromatosis" for consideration by *eLife*. Your article has been reviewed by 3 peer reviewers, including Beate Maria Lichtenberger as Reviewing Editor and Reviewer #1, and the evaluation has been overseen by Wafik El-Deiry as the Senior Editor. The following individual involved in review of your submission has agreed to reveal their identity: Charles Swanton (Reviewer #3).

Essential revisions:

1) The study presents novel findings and interesting results despite the small sample size of the experimental cohorts. However, few aspects of data visualization should be improved such as the graphical representation of differentially expressed genes. In addition, since this is a 4-generation HGF family and the sample size is large enough, genome-wide linkage analysis is essential

2) Several conclusions should be backed up by further experimental data or referencing previously published studies or removed:

– The suggestion that top down-regulated and up-regulated genes from RNA-seq lie downstream of ZNF862 and play a role in HGF pathophysiology.

– The conclusion that REST and ZNF862 may execute similar transcriptional functions

3) Please, acknowledge a previous body of work on ZNF862 in the manuscript

4) Data should be deposited so that readers can access them. The provided accession number the data cannot be found or accessed at the China Genebank Nucleotide Sequence Archive.

*Reviewer #1 (Recommendations for the authors):*

According to the authors the sequencing data set has been deposited in the China Genebank Nucleotide Sequence Archive. However, with the provided accession number the data cannot be found or accessed. The data should be deposited so that readers can access them.

It is not clear whether the data from 3 independent experiments shown in Figure 2B have been done with fibroblasts from the same donor or from different donors. If fibroblasts from only 1 donor were used, the number of biological replicates in figure 2B should be increased.

Could you provide data if shRNA knock-down of ZNF862 affects fibroblast proliferation?

Bulk RNA sequencing in Figure 2A indicates a down-regulation of SOS1 in HGF tissue. Since mutations in SOS1 have been associated with HGF, can you confirm that ZNF862 KD in gingival fibroblasts affects SOS1 expression?

The authors speculate that the top 100 up- or down-regulated genes lie downstream of ZNF862. However, they do not provide any evidence for this conclusion. If you cannot provide any data to support this conclusion, the statement should be down-tuned or removed.

Furthermore, pathway enrichment analysis revealed an association to several infection-related pathways. The authors conclude that this is due to an association with IL-6 signaling. However, differential expression of IL-6 or its downstream targets is not shown in the bulk sequencing, neither in the KD experiments. If the authors cannot provide any evidence, the statement should be removed.

*Reviewer #2 (Recommendations for the authors):*

Since this is a 4-generation HGF family and the sample size is large enough, genome-wide linkage analysis is essential. If the ZNF862 gene is located within the final linkage region, it would be a very important genetic evidence. Detecting the presence of the c.2812G>A variant in a certain number of normal controls is required.

*Reviewer #3 (Recommendations for the authors):*

Because of the novelty of the findings the work contributes to advancing the understanding of the genetic components of the disease. We recommend this article for publication in *eLife* after addressing revisions suggested below:

– Please acknowledge previously published work suggesting a plausible role of ZNF862 in matrix-producing metaplastic carcinoma (Schwartz et al., 2019) and association with IgE-mediated type-I hypersensitivity in children (Peng et al., 2018).

– Line 121 – the statement on ZNF862 ubiquitous expression should include a reference.

– Figure 2D – Please clarify what is meant by "gene number" in the legend of the graph and check if the decimal point is in the correct place. Currently it reads as though enrichment pathway is based on only 1 or 2 genes.

– Figure 2D – Please include two colours in the p-value scale to help the reader identify categories with statistical significance.

– Figure 2D – Please order categories either according to the p-value or to the enrichment for easier interpretation.

– Line 138 – Please specify in the text how many patients and controls were included in the RNA-seq experiment

– Line 175-181 – the statement that REST and ZNF862 might have similar mechanisms of action in causing HGF only because both have zinc finger domains in their structures is too speculative. Without referenced studies on REST downstream gene targets, suggested mechanism, comparison between REST and ZNF862 zinc finger DNA binding domains structures, suggestions of which protein complexes REST and ZNF862 might be part of, or any other experimental data this statement is not justified. This section requires more experimental evidence to support the connection between REST and ZNF862.

– Line 168 – The statement "These DE genes probably lie in the downstream of the ZNF862 and correlated with the HGF trait" is not supported by experimental data in the paper. To make this claim a ChIP-seq showing ZNF862 binding to those target genes, or qPCR validation of the finding in a larger cohort of patients with HGF, or functional studies demonstrating up and down regulations of those genes following ZNF862 knock down/knock out, or discussion of the genes function is required before publication.

– Line 277 – please change shRNA interfering for shRNA interference and include information about shRNA delivery into the cells in this section.

---

## [Author Response]

Reviewer #1 (Recommendations for the authors):According to the authors the sequencing data set has been deposited in the China Genebank Nucleotide Sequence Archive. However, with the provided accession number the data cannot be found or accessed. The data should be deposited so that readers can access them.

The WES and RNA sequencing data supporting this study have been deposited in the China Genebank Nucleotide Sequence Archive (https://db.cngb.org/cnsa, accession number CNP0000995), and the data now are totally publicly accessed. Thanks for the suggestion.

It is not clear whether the data from 3 independent experiments shown in Figure 2B have been done with fibroblasts from the same donor or from different donors. If fibroblasts from only 1 donor were used, the number of biological replicates in figure 2B should be increased.

We used the control cells from three different donors, and we added this information in the ShRNA interference section of the Materials and methods part of the revised manuscript. Thanks for the suggestion.

Could you provide data if shRNA knock-down of ZNF862 affects fibroblast proliferation?

We added this information in the Proliferation assay section in the Materials and methods part of the revised manuscript, and the figure is provided as Supplementary Figure 2—figure supplement 2. Thanks for the suggestion.

Actually, we did not observe a significant change of proliferation between the scrambler and shRNA groups during 5 days cell culture, which may be resulting from the too short time course to observe the sophisticated physiological effect.

Bulk RNA sequencing in Figure 2A indicates a down-regulation of SOS1 in HGF tissue. Since mutations in SOS1 have been associated with HGF, can you confirm that ZNF862 KD in gingival fibroblasts affects SOS1 expression?

We added this information as Supplementary Figure S3 of the revised manuscript. Thanks for the suggestion.

The authors speculate that the top 100 up- or down-regulated genes lie downstream of ZNF862. However, they do not provide any evidence for this conclusion. If you cannot provide any data to support this conclusion, the statement should be down-tuned or removed.

We changed the description “These DE genes probably lie in the downstream of the ZNF862 and correlated with the HGF trait.” into “These DE genes probably correlated with the HGF trait.” in the revised manuscript. Thanks for the suggestion.

Furthermore, pathway enrichment analysis revealed an association to several infection-related pathways. The authors conclude that this is due to an association with IL-6 signaling. However, differential expression of IL-6 or its downstream targets is not shown in the bulk sequencing, neither in the KD experiments. If the authors cannot provide any evidence, the statement should be removed.

We removed this description in the revised manuscript. Thanks for the suggestion.

Reviewer #2 (Recommendations for the authors):Since this is a 4-generation HGF family and the sample size is large enough, genome-wide linkage analysis is essential. If the ZNF862 gene is located within the final linkage region, it would be a very important genetic evidence. Detecting the presence of the c.2812G>A variant in a certain number of normal controls is required.

We really appreciate your valuable comments. However, we have some difficulties to achieve this. Since we only performed WES on 10 individuals in this family (the red color indicated in Figure 1A), the other members do not agree to provide additional DNA for WES sequencing at this moment, as a result, the genome-wide linkage analysis seems not easily achieved in our current situation.

Meanwhile, we calculated the LOD score of the ZNF862 mutation, LOD=3.6, that may be a statistical significance for a linkage evidence; considering the meiotic recombination may happen freely between haplotypes, this ZNF862 mutation may not be ruled out as the causative mutation in this family even if the ZNF862 mutation be not located in the significant region of genome-wide linkage analysis based on SNP markers. Our opinion is just for your reference. Thanks for the suggestion.

Reviewer #3 (Recommendations for the authors):Because of the novelty of the findings the work contributes to advancing the understanding of the genetic components of the disease. We recommend this article for publication in eLife after addressing revisions suggested below:– Please acknowledge previously published work suggesting a plausible role of ZNF862 in matrix-producing metaplastic carcinoma (Schwartz et al., 2019) and association with IgE-mediated type-I hypersensitivity in children (Peng et al., 2018).

We added this information in line 124 of the revised manuscript. Thanks for the suggestion.

– Line 121 – the statement on ZNF862 ubiquitous expression should include a reference.

We added the reference in the revised manuscript. Thanks for the suggestion.

– Figure 2D – Please clarify what is meant by "gene number" in the legend of the graph and check if the decimal point is in the correct place. Currently it reads as though enrichment pathway is based on only 1 or 2 genes.

We carefully revised the Figure 2D, and the gene number indicate the differentially expressed genes among the relevant pathway, we revised the legend of the graph in the Figure 2D. Thanks for the suggestion.

– Figure 2D – Please include two colours in the p-value scale to help the reader identify categories with statistical significance.

We carefully revised the Figure 2D. Thanks for the suggestion.

– Figure 2D – Please order categories either according to the p-value or to the enrichment for easier interpretation.

We carefully revised the Figure 2D. We ordered categories according to the enrichment. Thanks for the suggestion.

– Line 138 – Please specify in the text how many patients and controls were included in the RNA-seq experiment

We added this information in the revised manuscript. Totally two patients (IV-1 and IV-2) and four controls were included in the RNA-seq experiments. Thanks for the suggestion.

Additionally, in the Transcriptome sequencing section of the Materials and methods part of the revised manuscript. The details of the controls were described as following:

“The primary fibroblasts were isolated from fibromatic gingival specimens obtained from the patients who underwent gingivectomy (IV-1 as patient 1 and IV-2 as patient 2) using standard explant culture, while control cells were obtained from independent age- and gender-matched controls (two males at 25 and 26 years old as control 1 and 2 respectively, and two females at 21 and 23 years old as control 3 and 4 respectively) who underwent crown lengthening surgery for the restorative purpose.”

– Line 175-181 – the statement that REST and ZNF862 might have similar mechanisms of action in causing HGF only because both have zinc finger domains in their structures is too speculative. Without referenced studies on REST downstream gene targets, suggested mechanism, comparison between REST and ZNF862 zinc finger DNA binding domains structures, suggestions of which protein complexes REST and ZNF862 might be part of, or any other experimental data this statement is not justified. This section requires more experimental evidence to support the connection between REST and ZNF862.

We removed this REST mechanisms description in line 180-182 of the revised manuscript. Thanks for the suggestion.

– Line 168 – The statement "These DE genes probably lie in the downstream of the ZNF862 and correlated with the HGF trait" is not supported by experimental data in the paper. To make this claim a ChIP-seq showing ZNF862 binding to those target genes, or qPCR validation of the finding in a larger cohort of patients with HGF, or functional studies demonstrating up and down regulations of those genes following ZNF862 knock down/knock out, or discussion of the genes function is required before publication.

We changed the description “These DE genes probably lie in the downstream of the ZNF862 and correlated with the HGF trait.” into “These DE genes probably correlated with the HGF trait.” in the revised manuscript. Thanks for the suggestion.

– Line 277 – please change shRNA interfering for shRNA interference and include information about shRNA delivery into the cells in this section.

We revised the description in this section of the revised manuscript. Thanks for the suggestion.